# Estimating Epistemic and Aleatoric Uncertainty with a Single Model

**Matthew A. Chan**
Department of Computer Science
University of Maryland
College Park, MD 20742
mattchan@umd.edu

**Maria J. Molina**
Department of Atmospheric and Oceanic Science
University of Maryland
College Park, MD 20742
mjmolina@umd.edu

**Christopher A. Metzler**
Department of Computer Science
University of Maryland
College Park, MD 20742
metzler@umd.edu

## Abstract

Estimating and disentangling epistemic uncertainty, *uncertainty that is reducible with more training data*, and aleatoric uncertainty, *uncertainty that is inherent to the task at hand*, is critically important when applying machine learning to high-stakes applications such as medical imaging and weather forecasting. Conditional diffusion models' breakthrough ability to accurately and efficiently sample from the posterior distribution of a dataset now makes uncertainty estimation conceptually straightforward: One need only train and sample from a large ensemble of diffusion models. Unfortunately, training such an ensemble becomes computationally intractable as the complexity of the model architecture grows. In this work we introduce a new approach to ensembling, *hyper-diffusion models (HyperDM)*, which allows one to accurately estimate both epistemic and aleatoric uncertainty with a single model. Unlike existing single-model uncertainty methods like Monte-Carlo dropout and Bayesian neural networks, HyperDM offers prediction accuracy on par with, and in some cases superior to, multi-model ensembles. Furthermore, our proposed approach scales to modern network architectures such as Attention U-Net and yields more accurate uncertainty estimates compared to existing methods. We validate our method on two distinct real-world tasks: x-ray computed tomography reconstruction and weather temperature forecasting. Source code is publicly available at https://github.com/matthewachan/hyperdm.

## 1   Introduction

Machine learning (ML) based inference and prediction algorithms are being actively adopted in a range of high-stakes scientific and medical applications: ML is already deployed within modern computed tomography (CT) scanners [10], ML is actively used to search for new medicines [29], and over the last year ML has begun to compete with state-of-the-art weather and climate forecasting systems [46, 36, 7]. In mission-critical tasks like weather forecasting and medical imaging/diagnosis, the importance of reliable predictions cannot be overstated. The consequences of erroneous decisions in these domains can range from massive financial costs to, more critically, the loss of human lives. In this context, understanding and quantifying uncertainty is a pivotal step towards improving the robustness and reliability of ML models.

38th Conference on Neural Information Processing Systems (NeurIPS 2024).

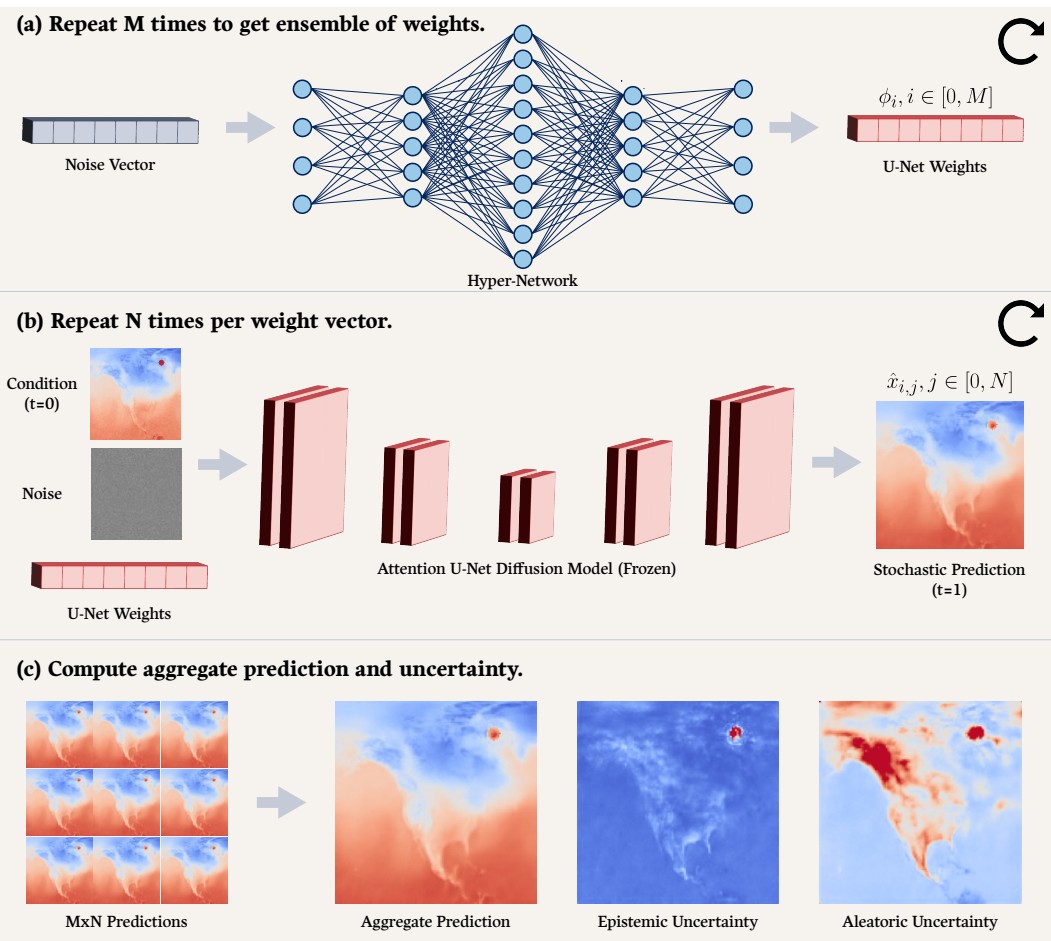

**(a) Repeat M times to get ensemble of weights.**

Noise Vector

Hyper-Network

$\phi_i, i \in [0, M]$

U-Net Weights

**(b) Repeat N times per weight vector.**

Condition (t=0)

Noise

U-Net Weights

Attention U-Net Diffusion Model (Frozen)

$\hat{x}_{i,j}, j \in [0, N]$

Stochastic Prediction (t=1)

**(c) Compute aggregate prediction and uncertainty.**

MxN Predictions

Aggregate Prediction

Epistemic Uncertainty

Aleatoric Uncertainty

Figure 1: **General framework of HyperDM.** (a) A Bayesian hyper-network is optimized to generate diffusion model weights from randomly sampled noise. This process is repeated $M$ times to obtain an ensemble of $M$ weights. (b) A diffusion model accepts fixed weights from the hyper-network to stochastically generate a prediction. This process is repeated $N$ times for each set of weights, yielding a total of $M \times N$ predictions. (c) The ensemble predictions are aggregated to produce a final prediction and an epistemic / aleatoric uncertainty map.

For an uncertainty estimate to be most useful, it must differentiate between *aleatoric* and *epistemic* uncertainty. Aleatoric uncertainty describes the fundamental variability and ill-posedness of the inference task. By contrast, epistemic uncertainty describes the inference model's lack of knowledge or understanding—which can be reduced with more diverse training data. Distinguishing between these two types of uncertainty provides valuable insights into the strengths and weaknesses of a predictive model, offering pathways towards improving its performance. In applications like weather forecasting, epistemic uncertainty can be used to inform the optimal placement of new weather stations. Additionally, in medical imaging, decomposition of uncertainty into its aleatoric and epistemic components is important for identifying out-of-distribution measurements where model predictions should be verified by trained experts.

This work presents a new approach for estimating aleatoric and epistemic uncertainty *using a single model*. Specifically, our approach uses a novel pipeline integrating a conditional diffusion model [23] and a Bayesian hyper-network [34] to generate an ensemble of predictions. Conditional diffusion models allow one to sample from an implicit representation of the posterior distribution of an inverse problem. Meanwhile, hyper-networks allow one to sample over a collection of networks that are consistent with the training data. Together, these components can efficiently estimate both sources of uncertainty, without sacrificing inference accuracy. Our specific contributions are summarized below:

Table 1: **Comparison of training and inference times.** The time required to train an $M = 10$ member ensemble on the LUNA16 dataset is shown in the second column. The third column shows the time required to generate a predictive distribution of size $M \times N = 1000$ for a single input.

| METHOD | TRAINING TIME (MINUTES) | EVALUATION TIME (MINUTES) |
|---|---|---|
| MC-DROPOUT [18] | 47.03 | 3.70 |
| DPS-UQ [14] | 441.09 | 3.31 |
| HYPERDM | 48.53 | 3.18 |

- We apply Bayesian hyper-networks in a novel setting (i.e., diffusion models) to estimate both epistemic and aleatoric uncertainties from a single model.

- We conduct a toy experiment with ground truth uncertainties and show that the proposed method accurately predicts both sources of uncertainty.

- We apply the proposed method on two mission-critical real-world tasks, CT reconstruction and weather forecasting, and demonstrate that our method achieves a significantly lower training overhead and better reconstruction quality compared to existing methods.

- We conduct ablation studies investigating the effects of ensemble size and the number of ensemble predictions on uncertainty quality, which show (i) that larger ensembles improve out-of-distribution detection and (ii) that additional predictions smooth out irregularities in aleatoric uncertainty estimates.

## 2 Related Work

### 2.1 Uncertainty Quantification

Probabilistic methods are commonly used to estimate uncertainty by first generating an ensemble of models and subsequently quantifying uncertainty as the variance or entropy over the ensemble's predictions [11]. Deep ensembles [35] explicitly train such an ensemble to predict epistemic uncertainty. However, with modern neural network architectures exceeding a billion parameters, the computational cost required to train deep ensembles is prohibitively expensive.

Other methods attempt to approximate deep ensembles while circumventing its training overhead. Bayesian neural networks (BNNs) [40, 44] use variational inference [19, 65] to model the posterior weight distribution. Monte-Carlo (MC) dropout [18] leverages dropout [57] to stochastically induce variability in the network's predictions. Recent works in weather forecasting [7, 46] perturb network inputs with random noise to similarly generate stochastic predictions. Still, each of these methods has notable trade-offs preventing their widespread adoption. BNNs incur a runtime cost that scales proportionally with the number of model parameters, leading to slow inference and training times. MC dropout and input perturbation introduce noise into the inference process, which adversely affects model prediction quality. Moreover, perturbing inputs with noise is not equivalent to deep ensembles (in the Bayesian sense) as these methods optimize the weights of a single, deterministic model.

A separate branch of research [48, 3] explores distribution-free uncertainty estimation, which uses conformal prediction [4] and quantile regression [32] to estimate bounds on aleatoric uncertainty. Subsequent works [58] have extended this to additionally estimate epistemic uncertainty; however, these methods use deep ensembles to do so—leading to similar issues with computational complexity.

### 2.2 Hyper-Networks

Hyper-networks [20] employ a unique paradigm where one network—the "hyper-network"—generates weights for another "primary" network. This framework circumvents the need to train multiple task-specific or dataset-specific models. Instead, one need only train a single hyper-network to cover a range of tasks or datasets. Given an input token representing a specific task, a hyper-network learns to generate reasonable weights with which the primary network can accomplish that task [61]. Note that, during training, losses are back-propagated such that only the hyper-network's parameters are updated while the primary network's weights are purely generated by the hyper-network.

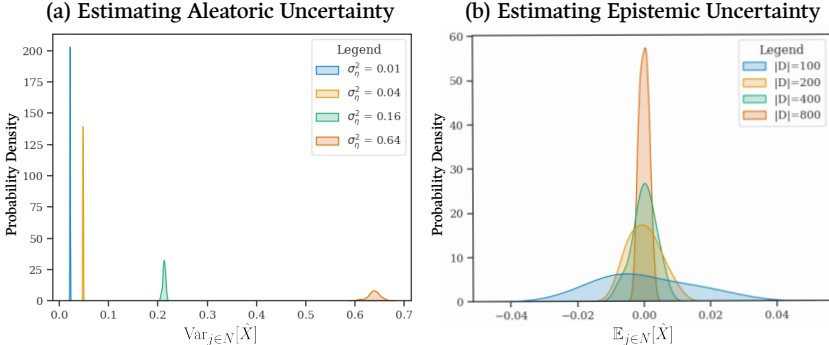

Figure 2: **Accurate uncertainty estimation using HyperDM.** (a) HyperDM is trained on four 1D datasets with aleatoric uncertainty determined by noise variance $\sigma_\eta^2$. Variances across diffusion model predictions are visualized as one distribution per training dataset. Aleatoric estimates (i.e., the mean of each distribution) accurately predict $\sigma_\eta^2$. (b) HyperDM is trained on four datasets with epistemic uncertainty determined by dataset size $|\mathcal{D}|$. Prediction means are visualized as one distribution per training dataset. Epistemic estimates (i.e., the variance of each distribution) grow inversely with $|\mathcal{D}|$.

Bayesian hyper-networks (BHNs) [34, 33] extend hyper-networks to quantify uncertainty. Rather than accepting task-specific tokens as inputs, BHNs accept random noise and stochastically generate weights for the primary network. BHNs thus serve as an implicit representation for the true posterior weight distribution [47, 28]. Epistemic uncertainty is measured as the variance across predictions yielded by the primary network for different weights sampled from the BHN.

## 2.3 Diffusion Models

Diffusion models (DMs) [52, 55, 56] represent a class of generative machine learning models that learns to sample from a target distribution. These models fit to the Stein score function [39] of the target distribution by iteratively transitioning between an easy-to-sample (typically Gaussian) distribution and the target distribution. During training, samples from the target distribution are corrupted by running the forward "noising" diffusion process, and the network learns to estimate the added noise. To generate samples, the network iteratively denoises images of pure noise until they looks like they were sampled from the target distribution [23]. DMs have shown success in generating high-quality, realistic images and capturing diverse data distributions [12]. To date, however, there has been limited research [6] investigating the use of DMs for uncertainty estimation.

## 3 Problem Definition

Given measurements $y \sim \mathcal{Y}$ corresponding to signals of interest $x \sim \mathcal{X}$, our objective is to train a model which can simultaneously recover $x$ and quantify the aleatoric and epistemic uncertainty of its predictions. The predictive distribution of a such a model is given by

$$p(x|y, \mathcal{D}) = \int p(x|y, \phi)p(\phi|\mathcal{D})d\phi \tag{1}$$

where $p(x|y, \phi)$ is the likelihood function, and $p(\phi|\mathcal{D})$ is the posterior over model parameters $\phi$ for a training dataset $\mathcal{D}$ [59]. Uncertainty on this distribution stems from two distinct sources: aleatoric uncertainty and epistemic uncertainty [13].

### 3.1 Aleatoric Uncertainty

Aleatoric uncertainty (AU) arises from inherent randomness in the underlying measurement process and is represented by the likelihood function in Equation (1). Most notably, this source of uncertainty is irreducible for a given measurement process [17, 30]. In the context of predictive modeling, AU represents how ill-posed the task is and is often associated with noise, measurement errors, or inherent unpredictability in the observed phenomena.

Consider an inverse problem where the goal is to recover $x$ from measurements

$$y = \mathcal{F}(x) + \eta, \tag{2}$$

defined by forward operator $\mathcal{F}$ and non-zero measurement noise $\eta \sim \mathcal{N}(0, \sigma^2)$, by learning the inverse mapping $\mathcal{F}^{-1} : \mathcal{Y} \to \mathcal{X}$. Even with a perfect model capable of sampling from the true likelihood $p(x|y)$, irreducible errors are still present due to the ambiguity $\eta$ around which $x \sim p(x)$ explains the observed measurement $y$. This ambiguity captures the inherent randomness (i.e., the *aleatoric uncertainty*) of the inverse problem and is measured by the variance $\sigma^2$ of $\eta$ [27].

## 3.2 Epistemic Uncertainty

Epistemic uncertainty (EU) relates to a lack of knowledge or incomplete understanding of a problem and is reducible with additional training data [27]. This type of uncertainty reflects limitations in a model's knowledge and its ability to accurately capture underlying patterns in the data. Assume we initialize $M$ models with random weights $\{\phi_i\}_{i=0}^{M}$ and sufficient capacity to perfectly capture the inverse model described in Section 3.1. After training, discrepancies (i.e., *epistemic uncertainty*) inevitably arise in the final weights learned by each model, due to the random weight initialization process. As additional training data is provided, model weights converge more strongly—corresponding to a reduction in EU [11, 27].

# 4 Method

We measure uncertainty using variance and apply the law of total variance [11, 60, 50] to decompose total uncertainty (TU) across model predictions $\hat{X} \sim p(x|y, \mathcal{D})$ into its AU and EU components

$$\mathrm{Var}(\hat{X}) = \underbrace{\mathrm{Var}_{\phi \sim p(\phi|\mathcal{D})}\left[\mathbb{E}_{\hat{x} \sim p(x|y, \phi)}\left[\hat{X}\right]\right]}_{\text{EU}} + \underbrace{\mathbb{E}_{\phi \sim p(\phi|\mathcal{D})}\left[\mathrm{Var}_{\hat{x} \sim p(x|y, \phi)}\left[\hat{X}\right]\right]}_{\text{AU}}. \tag{3}$$

The first term captures the explainable uncertainty, given by the variance of sampled weights $\phi \sim p(\phi|\mathcal{D})$ over the expected values of samples $\hat{x} \sim p(x|y, \phi)$ from the likelihood function. This term ignores variance caused by the ill-posedness of the likelihood function and therefore represents EU. The second term captures the unexplainable uncertainty and is given by the expectation of sampled weights $\phi \sim p(\phi|\mathcal{D})$ over the variance of samples $\hat{x} \sim p(x|y, \phi)$ from the likelihood function. This term ignores variance caused by the sampling of weights from the posterior and therefore represents AU.

Both the likelihood function $p(\phi|\mathcal{D})$ and the posterior $p(x|y, \phi)$ do not have an explicit closed-form, making computation of (3) intractable. To circumvent this, we instead learn their respective implicit distributions [28, 47] $q(\phi)$ and $q(x|y)$.

## 4.1 Implicit Likelihood Function

As demonstrated in [55], DMs enable sampling from an implicit conditional distribution $q(x|y)$ by learning to invert a diffusion process that gradually transforms a target data distribution into a simple (typically Gaussian) data distribution [56]. The forward diffusion process can be described by a $T$ length Markov chain

$$q(x^{(t)}|x^{(t-1)}) := \mathcal{N}\left(x^{(t)}; \sqrt{1 - \sigma_t^2} x^{(t-1)}, \sigma_t^2\right) \tag{4}$$

that transforms samples $x^{(0)}$ from the data distribution into samples $x^{(T)}$ from a Gaussian distribution [23]. Conversely, the reverse diffusion process

$$p(x^{(t-1)}|x^{(t)}, y) := \mathcal{N}(x^{(t-1)}; x^{(t)} + \sigma_t^2 \nabla_{x^{(t)}} \log p(x^{(t)}|y), \sigma_t^2) \tag{5}$$

transforms pure noise into samples from $p(x|y)$ [63].

Since explicit computation of the score function $\nabla_{x_t} \log p(x^{(t)}|y)$ is intractable [16], a neural network $s(x, t|y, \phi)$ is typically trained to approximate it via an L2 minimization objective

$$\mathbb{E}_{(x,y)\sim\mathcal{D}}\left[\left\|\nabla_{x^{(t)}} \log p(x^{(t)}|y) - s(x^{(t)}, t|y, \phi)\right\|_2^2\right] \tag{6}$$

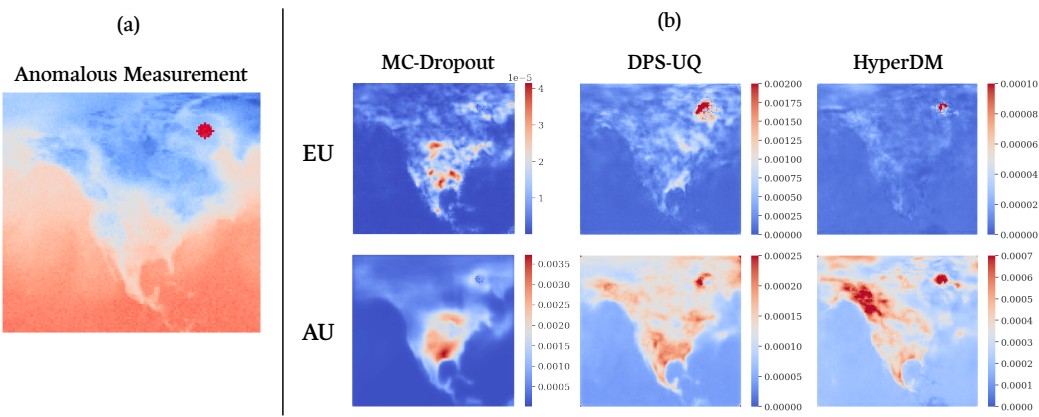

Figure 3: **Weather forecasting on out-of-distribution data.** (a) An out-of-distribution measurement is formed by synthetically inserting a hot spot in the northeastern part of Canada. (b) Epistemic and aleatoric uncertainty maps are produced by each method on the provided measurement. Compared to other methods, HyperDM is best able to isolate the abnormal feature in its epistemic estimate.

where $\mathcal{D} = \{(x_0, y_0), \ldots, (x_N, y_N)\}$ represents the training dataset. Once the DM has finished training, we can sample from the implicit likelihood function $q(x|y)$ by first sampling random noise $x^{(T)} \sim \mathcal{N}(0, \sigma^2)$ and iteratively denoising the image $T$ times following Equation (5) to obtain $x^{(0)} \sim q(x|y)$ [42].

## 4.2 Implicit Posterior Distribution

Similar to the likelihood function, the posterior weight distribution has no explicit closed-form representation, so we instead make use of an implicit distribution $q(\phi)$ to approximate $p(\phi|\mathcal{D})$. As mentioned in Section 2.2, BHNs [34] enable sampling from $q(\phi)$ by transforming samples $z \sim \mathcal{N}(0, \sigma^2)$ into weights $\phi \sim q(\phi)$ for the primary network. In the case of the inverse imaging problem from Section 3.1, the primary network would be a network $f(\cdot|\phi)$ with parameters $\phi$ that learns the inverse mapping from measurements to signals $\mathcal{Y} \rightarrow \mathcal{X}$.

Training a BHN differs from conventional deep learning methods in that the weights of the primary network $\phi$ are generated by a hyper-network and are thus not learnable parameters. Instead, the weights $\theta$ of a BHN $h_\theta$ are optimized via the minimization objective

$$\mathbb{E}_{(x,y)\sim\mathcal{D}, z\sim\mathcal{N}(0,\sigma^2)} \left[ \|f(y \mid h_\theta(z)) - x\|_2^2 \right] \tag{7}$$

where $h_\theta$ maps random input vectors $z \sim \mathcal{N}(0, \sigma^2)$ to weights $\phi$ (see Figure 1a). Importantly, weights produced by the BHN do not collapse to a mode because there are many network weights which yield plausible predictions with respect to L2 distance. As less data is available during training, a broader range of network weights reasonably explain that data.

## 4.3 Estimation of Aleatoric and Epistemic Uncertainty with a Single Model

We leverage DMs and BHNs to implicitly model $p(x|y, \phi)$ and $p(\phi|\mathcal{D})$, respectively, thus enabling sampling from both distributions. Specifically, our framework consists of a BHN $h_\theta$ that generates weights $\phi_i \sim q(\phi)$ for a DM $s(\cdot|\phi)$, which we collectively refer to as a hyper-diffusion model (HyperDM). At inference time, we sample $i \in M$ weights from $h_\theta$ and for each weight $\phi_i$ generate $j \in N$ samples from $s(\cdot|\phi_i)$—yielding a distribution of $M \times N$ predictions $\hat{x}_{i,j}$ (see Figure 1). This framework is a transformation of Equation (1), where both posterior and likelihood have been replaced with implicit distributions to yield a tractable approximation of the predictive distribution

$$p(x|y, \mathcal{D}) \approx \int q(x|y)q(\phi)d\phi. \tag{8}$$

Table 2: **Ensemble prediction quality on real-world data.** Baseline image quality assessment scores are calculated on test data from a CT dataset (i.e., LUNA16) and a weather forecasting dataset (i.e., ERA5). Best scores are highlighted in red and second best scores are highlighted in blue.

| | LUNA16 | | | ERA5 | | |
|---|---|---|---|---|---|---|
| METHOD | SSIM ↑ | PSNR (DB) ↑ | CRPS ↓ | SSIM ↑ | PSNR (DB) ↑ | CRPS ↓ |
| MC-DROPOUT [18] | 0.77 | 30.25 | 0.023 | 0.93 | 31.34 | 0.034 |
| DPS-UQ [14] | 0.89 | 34.95 | 0.01 | 0.94 | 32.83 | 0.013 |
| HYPERDM | 0.87 | 35.16 | 0.01 | 0.95 | 33.15 | 0.012 |

Applying Equation (3), uncertainty over the predictive distribution $\hat{X} = \{\hat{x}_{i,j}, \ldots, \hat{x}_{M,N}\}$ is decomposed into its respective aleatoric and epistemic components,

$$\widehat{\text{AU}} = \mathbb{E}_{i \in M} \left[ \text{Var}_{j \in N} \left[ \hat{X} \right] \right] \tag{9}$$

$$\widehat{\text{EU}} = \text{Var}_{i \in M} \left[ \mathbb{E}_{j \in N} \left[ \hat{X} \right] \right], \tag{10}$$

such that $\widehat{\text{TU}} = \widehat{\text{AU}} + \widehat{\text{EU}}$. Following existing ensemble methods [35, 46, 7], we compute the aggregate ensemble prediction as the expectation over $\hat{X}$, formally expressed as

$$\mathbb{E}_{i \in M, j \in N} \left[ \hat{X} \right]. \tag{11}$$

Compared to other aggregation methods (e.g., median, mode), we observe the best performance when taking the ensemble mean. Please refer to Figure 5 and Table 3 in the supplement for more details.

Unlike deep ensembles which require training $M$ distinct models to compute EU and AU, HyperDM only requires training a single model (i.e., a BHN)—theoretically consuming up to $M$-fold fewer computational resources. Furthermore, unlike many pseudo-ensembling methods [18, 7, 46], HyperDM doesn't need to exploit randomness caused by perturbations to model $p(\phi|\mathcal{D})$—avoiding adverse effects on model performance. Moreover, unlike BNNs, HyperDM is relatively cheap to sample from in terms of computational runtime and resources—making it significantly faster and more scalable compared to BNN-based uncertainty estimation methods.

Differing from prior work [60], we make no Gaussian assumptions on the predictive distribution $p(x|y, \mathcal{D})$ nor on the likelihood function $p(x|y, \phi)$. This is because our method approximates $p(x|y, \phi)$ by repeatedly sampling $\hat{x} \sim q(x|y)$ from a DM, rather than explicitly modeling the distribution as a Gaussian with mean $\mu$ and variance $\sigma^2$. Therefore, our aggregate predictive distribution $p(x|y, \mathcal{D})$ is not restricted to a Gaussian mixture model $\mathcal{N}(\mu_*(x), \sigma_*^2(x))$ over the collective mean $\mu_*$ and variance $\sigma_*^2$ of all ensemble members.

## 5 Experiments

Please refer to Appendix A for training details (e.g., network architectures and loss functions).

**Baselines.** We focus on comparing HyperDM against methods which are similarly capable of estimating both EU and AU. Our benchmark consists of a state-of-the-art method, deep-posterior sampling for uncertainty quantification [14] (henceforth referred to as DPS-UQ), and a dropout-based method (referred to as MC-Dropout). DPS-UQ is implemented as an $M$-member ensemble of deep-posterior sampling (DPS) DMs. MC-Dropout is implemented as a single DM with weights sampled from $q(\phi)$ using dropout instead of a BHN. Despite its inability to jointly predict EU and AU, we also include a BNN baseline in our initial experiments to illustrate the advantages of our method in terms of prediction speed and accuracy.

**Metrics.** We evaluate the quality of baseline predictions using both full-reference image quality metrics and distribution-based metrics. Specifically, we compute peak signal-to-noise ratio (PSNR) [24] and structural similarity index (SSIM) [62] between mean predictions (see Equation (11)) and their corresponding ground truth references. We also compute the continuous ranked probability score (CRPS) [41] as a holistic indicator of the quality of $\hat{X}$, given by

$$\text{CRPS}(F, a) = \int_{-\infty}^{\infty} \left[ F(a) - \mathbb{1}_{a \geq x} \right]^2 da, \tag{12}$$

where $F$ is the cumulative distribution function of $\hat{X}$ and $\mathbb{1}$ is the Heaviside step function.

In our initial experiments, we compare baseline estimates of AU and EU against ground truth uncertainty. However, extending such validation to more complex tasks and datasets is difficult because uncertainty is affected by a wide variety of environmental factors (e.g., measurement noise, sampling rates) which are often unreported. As a result, in subsequent experiments, we follow [14] and evaluate uncertainty by generating out-of-distribution (OOD) measurements and verifying whether baseline estimates of $\widehat{\text{EU}}$ correctly predict OOD pixels.

## 5.1 Toy Problem

We first evaluate our method on a toy inverse problem to establish the correctness of our uncertainty estimates under a simple forward model where ground truth uncertainty is explicitly quantifiable. Training datasets are generated using the function

$$x = \sin(y) + \eta \tag{13}$$

where $\eta \sim \mathcal{N}(0, \sigma_\eta^2)$ and measurements $y \sim \mathcal{U}(-5, 5)$. We conduct two separate experiments to validate our method's ability to estimate uncertainty against ground-truth EU and AU.

**Estimating AU.** To test our method's ability to estimate AU, we generate four training datasets using Equation (13) with ground truth AU characterized by noise variances $\sigma_\eta^2 \in \{0.01, 0.04, 0.16, 0.64\}$. Each training dataset has $|\mathcal{D}| = 500$ examples, and a HyperDM is trained on each dataset for 500 epochs. After training, we sample $M = 10$ weights from $h_\theta$ and $N = 10000$ realizations from $s(\cdot|\phi)$ to obtain a distribution of $M \times N$ predictions. We compute $\widehat{\text{AU}}$ for each ensemble member using Equation (9) and visualize $\text{Var}_{j \in N}[\hat{X}]$ across all $M$ weights in Figure 2a. The AU estimates across the four datasets are $\widehat{\text{AU}} = \{0.02, 0.05, 0.21, 0.64\}$, which closely match the ground-truth.

**Estimating EU.** We test our method's ability to estimate EU by generating four training datasets of varying sizes $|\mathcal{D}| \in \{100, 200, 400, 800\}$ and fixed noise variance $\sigma_\eta^2 = 0.01$. Unlike AU, ground truth EU cannot be explicitly quantified because it is independent from the training data [5]. As a result, we follow prior works [35, 13, 38] and validate EU qualitatively. We train a HyperDM on each dataset for 500 epochs and draw $M \times N$ samples from it where $M = 10, N = 10000$. We then calculate $\widehat{\text{EU}}$ using Equation (10) and plot $\mathbb{E}_{j \in N}[\hat{X}]$ for all $M$ weights in Figure 2b. The EU estimates across the four datasets are $\widehat{\text{EU}} = \left\{1.92 \times 10^{-4}, 2.20 \times 10^{-5}, 1.17 \times 10^{-5}, 1.83 \times 10^{-6}\right\}$, ordered by increasing $|\mathcal{D}|$. As expected, $\widehat{\text{EU}}$ decreases as $|\mathcal{D}|$ grows increasingly larger.

To highlight the key advantages of HyperDM over traditional uncertainty estimation techniques, we also train a BNN on the same datasets and sample $M = 10000$ predictions. The resulting estimates $\widehat{\text{EU}} = \{0.091, 0.071, 0.050, 0.012\}$ indicate a similar inversely proportional relationship with $|\mathcal{D}|$. However, despite having the same backbone architecture and training hyper-parameters, we observe aggregate predictions of lower individual and mean quality from the BNN when compared to HyperDM (see Table 4 of the supplement). Moreover, we observe that BNNs take over $2\times$ longer to train compared to HyperDM (i.e., 70 seconds vs. 30 seconds), and inference is an order of magnitude slower (i.e., 8.7 seconds vs. 0.7 seconds to generate 10,000 predictions), due to BNN's need to sample from the weight distribution at runtime.

## 5.2 Computed Tomography

In this experiment, we demonstrate our method's applicability for medical imaging tasks, specifically CT reconstruction. Using the Lung Nodule Analysis 2016 (LUNA16) [51] dataset, we form a target image distribution $\mathcal{X}$ by extracting 1,200 CT images, applying $4\times$ pixel binning to produce $128 \times 128$ resolution images, and normalizing each image by mapping pixel values between $[-1000, 3000]$ Hounsfield units to the interval $[-1, 1]$. We subsequently compute the sparse Radon transform with 45 projected views and add Gaussian noise with variance $\sigma^2 = 0.16$ to the resulting sinograms. Using filtered back-projection (FBP) [25], we obtain low-quality reconstructions $\mathcal{Y}$ of original images $\mathcal{X}$. The dataset is finally split into a training dataset comprised of 1,000 image-measurement pairs and a validation dataset of 200 data pairs. Following the training procedure described in Appendix A, we train MC-Dropout, DPS-UQ, and HyperDM on LUNA16. For fair comparison, all baselines are sampled from using $M = 10$ and $N = 100$ for a total of $M \times N = 1000$ predictions. We refrain

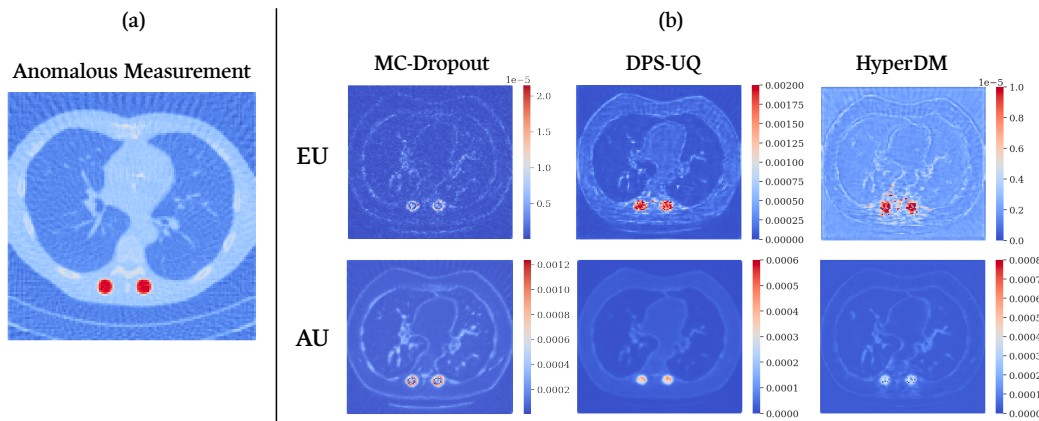

Figure 4: **CT reconstruction on out-of-distribution data.** (a) An out-of-distribution CT measurement formed by synthetically inserting metal implants along the spine. (b) Epistemic and aleatoric uncertainty maps are produced by each method on the out-of-distribution measurement. Both DPS-UQ and HyperDM are able to distinguish the abnormal feature in their epistemic prediction.

from training a BNN baseline on this dataset due to the high computational resources and runtime required to scale to the image domain.

In Table 2, we show average CRPS, PSNR, and SSIM scores computed over the test dataset. The relatively low image quality scores obtained by MC-Dropout are indicative of the adverse effects caused by randomly dropping network weights at inference time. Meanwhile, DPS-UQ reconstructions achieve a 15.5% higher average PSNR than MC-Dropout, but at the cost of an eight-fold increase in training time (see Table 1). On the other hand, HyperDM yields predictions of similar—and sometimes better—quality than DPS-UQ while only adding a 3% overhead in training time compared to MC-Dropout. Note that the discrepancy in training times between DPS-UQ and HyperDM will continually widen as we scale the ensemble size beyond $M = 10$. However, due to the high computational costs required to train $M > 10$ member deep ensembles, we limited baselines to ten-member ensembles for this experiment.

To evaluate the quality of baseline uncertainty predictions, we first select a random in-distribution image $x \in \mathcal{X}$ and generate its corresponding OOD measurement by first artificially inserting an abnormal feature (i.e., metal implants along the spinal column) and subsequently computing the corresponding FBP measurement $y$. Results in Figure 4 show that DPS-UQ and HyperDM yield comparable results in that their $\widehat{\text{EU}}$ predictions successfully highlight the OOD implant. In contrast, MC-Dropout fails to highlight OOD pixels in its $\widehat{\text{EU}}$ prediction. While prior work [43] suggests that AU estimates are unreliable—and should be subsequently disregarded—whenever EU is high, we nonetheless include $\widehat{\text{AU}}$ results in Figure 4 to demonstrate that HyperDM produces $\widehat{\text{AU}}$ predictions similar to that of a deep ensemble.

### 5.3 Weather Forecasting

In this experiment, we demonstrate the applicability of HyperDM for climate science—specifically two-meter surface temperature forecasting. Using the European Centre for Medium-Range Weather Forecasts Reanalysis v5 (ERA5) dataset [22], we generate a dataset comprised of 1,240 surface air temperature maps sampled at six-hour time intervals (i.e., $00, 06, 12, 18$ UTC) in January between 2009-2018. Images are binned down to $128 \times 128$ resolution and normalized such that pixel values between $[210, 313]$ Kelvin map to the interval $[-1, 1]$. Following experiments done in [46], we form data pairs $(x, y)$ using historical temperature data at time $t$ as the initial measurement image $y$ and data at time $t + 6$ hours as the target image $x$. A total of 200 images are held-out and used for validation and testing purposes.

Using the same training procedure as Section 5.2, we train MC-Dropout, DPS-UQ, and HyperDM on ERA5 and generate predictions with sampling rates $M = 10$ and $N = 100$. Baseline PSNR, SSIM,

and CRPS scores are reported in Table 2, where we observe trends similar to the prior experiment: DPS-UQ achieves a 5% higher average PSNR score compared to MC-Dropout, and HyperDM achieves 1% higher average PSNR score compared to DPS-UQ. Training overhead for DPS-UQ remains around $8\times$ that of MC-Dropout and HyperDM due to the need to repeat training $M$ times.

To generate OOD measurements, we first obtain an in-distribution measurement and subsequently insert an anomalous hot spot over northeastern Canada. Inspecting results in Figure 3, we observe that HyperDM's $\widehat{EU}$ prediction more accurately identifies OOD pixels than DPS-UQ. In contrast, MC-Dropout fails to identify the hot spot in its $\widehat{EU}$ prediction and instead incorrectly identifies regions in the central United States as OOD. Interestingly, all methods predict lower $\widehat{AU}$ over the ocean versus the North American continent, which aligns with our expectations, as water has less temperature variability compared to land due to its higher specific heat. Additional qualitative results showing the decomposition of $\widehat{TU}$ into its $\widehat{EU}$ and $\widehat{AU}$ components are provided in Figure 8 of the supplement.

## 6 Limitations and Future Work

We acknowledge two main limitations of our approach and identify potential avenues for improvement. Firstly, as a consequence of their iterative denoising process, inference on DMs is slow compared to inference on classical neural network architectures. However, recent advances in accelerated sampling strategies have largely mitigated this issue and allow for few [53] (and in some cases single [54]) step sampling from DMs. Secondly, hyper-networks suffer from a scalability problem in that their number of parameters scales with the number of primary network parameters. This stems from the fact that the dimensionality of the hyper-network's output layer is (in most cases) proportional to the number of parameters in the primary network [9]. Several works address this issue by proposing more efficient weight generation strategies [61, 2, 26]. Nonetheless, these problems remain a promising avenue for future research.

## 7 Conclusion

The growing application of ML to impactful scientific and medical problems has made accurate estimation of uncertainty more important than ever. Unfortunately, the gold standard for uncertainty estimation—deep ensembles—is prohibitively expensive to train, especially on modern network architectures containing billions of parameters. In this work, we propose HyperDM, a framework capable of approximating deep ensembles at a fraction of the computational training cost. Specifically, we combine Bayesian hyper-networks and diffusion models to generate a distribution of predictions with which we can estimate total uncertainty and its epistemic and aleatoric sub-components. Our experiments on weather forecasting and CT reconstruction demonstrate that HyperDM significantly outperforms pseudo-ensembling techniques like Bayesian neural networks and Monte Carlo dropout in terms of prediction quality. Moreover, when compared against deep ensembles, HyperDM achieves up to an $M\times$ reduction in training time while yielding predictions of similar (if not superior) quality, where $M$ is the ensemble size. This work thus makes a major stride towards developing accurate and scalable estimates of uncertainty.

## Acknowledgments and Disclosure of Funding

This work was supported in part by a University of Maryland Grand Challenges Seed Grant and NSF Award No. 2425735. M.A.C. and C.A.M. were supported in part by AFOSR Young Investigator Program Award No. FA9550-22-1-0208 and ONR Award No. N00014-23-1-2752.

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

# A   Training Details

All baselines are trained on a single NVIDIA RTX A6000 using a batch size of 32, an Adam [31] optimizer, and a learning rate of $1 \times 10^{-4}$. Training is run over 500 epochs in our initial experiment and 400 epochs in our CT and weather experiments. DMs are trained using a Markov chain of $T = 100$ timesteps.

## A.1   Network Architecture

The backbone architecture for all baselines (i.e., BNN, DPS-UQ, HyperDM) in the toy experiment from Section 5.1 is a multi-layer perceptron (MLP) [49] with five linear layers and rectified linear unit (ReLU) [1] activation functions. For experiments described in Sections 5.2 and 5.3, we scale the DM's backbone architecture up to an Attention U-Net [45] for all baselines. The U-Net consists of an initial 2D convolutional layer, followed by four $2\times$ downsampling ResNet [21] blocks, two middle ResNet blocks, four $2\times$ upsampling ResNet blocks, and a final 2D convolutional layer. Each ResNet block consists of two 2D convolutional layers—with group normalization [64] and Sigmoid Linear Units (SiLU) [15] activation function—as well an additional attention layer.

## A.2   Loss Functions

The training procedure for HyperDM is identical to that of a standard DM, except that the DM's weights are sampled from a BHN $h_\theta$. For each training pair $(x, y)$, we sample DM weights by first sampling random noise $z \sim \mathcal{N}(0, \sigma_z^2), z \in \mathbb{R}^8$ and then computing $\phi \sim h_\theta(z)$. We manually set the DM weights equal to $\phi$ and compute the loss function

$$\mathcal{L}_{\text{HyperDM}} = \|\epsilon - s(x^{(t)}, t|y, h_\theta(z))\|_2^2, \tag{14}$$

where $\epsilon \sim \mathcal{N}(0, \sigma^2)$ is the noise added to $x$ at time step $t$ and $s(\cdot|\phi)$ represents the DM. In general, we found HyperDM training to be stable across a variety of training hyper-parameters and did not encounter any over-fitting issues.

We follow [37] and train our BNN baseline $b(\cdot|\phi)$ by minimizing the loss function

$$\mathcal{L}_{\text{BNN}} = \|x - b(y|\phi)\|_2^2 + \lambda \text{KL}(q(\phi) \parallel p(\phi|\mathcal{D})), \tag{15}$$

which consists of a data fidelity term and an additional Kullback-Leibler (KL) divergence term between the true posterior $p(\phi|\mathcal{D})$ and the implicit distribution $q(\phi)$—approximated using Bayes by Backprop [8]. The weights of each BNN layer are sampled from a zero-mean normal distribution with standard deviation $\sigma = 0.1$, and the KL component of the loss term is down-weighted by $\lambda = 0.01$.

The training procedure for DPS-UQ and MC-Dropout are identical in that they share the same loss objective

$$\mathcal{L}_{\text{DPS-UQ}} = \mathcal{L}_{\text{MC-Dropout}} = \|\epsilon - s(x^{(t)}, t|y, \phi)\|_2^2, \tag{16}$$

where the weights $\phi$ are randomly initialized at the start of training and updated via backpropagation. However, we train $M$ separate DM instances for DPS-UQ, whereas only a single DM is trained for MC-Dropout.

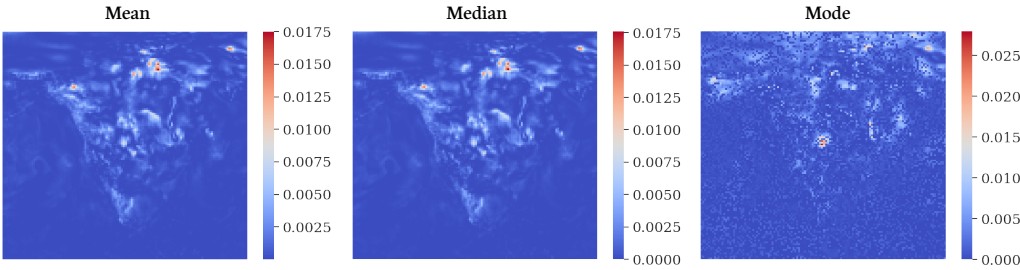

Figure 5: **Aggregation of ensemble predictions.** Ensemble predictions are aggregated using conventional methods (e.g., mean, median, mode). Mean and median aggregation results are similar, while mode aggregation results are noticeably more noisy.

Table 3: **Reconstruction quality of different ensemble aggregation methods.** HyperDM reconstruction results on the ERA5 test set are shown for three different ensemble aggregation strategies: mean, median, and mode. Best scores are highlighted in red and second best scores are highlighted in blue.

| AGGREGATION | SSIM $\uparrow$ | PSNR (DB) $\uparrow$ | L1 $\downarrow$ | CRPS $\downarrow$ |
|---|---|---|---|---|
| MEAN | 0.9455 | 32.93 | 0.018 | 0.01292 |
| MEDIAN | 0.9452 | 33.06 | 0.017 | 0.01294 |
| MODE | 0.6690 | 25.54 | 0.044 | 0.01293 |

To build a predictive distribution of size $M \times N$ with MC-Dropout, we first seed a pseudo-random number generator (RNG), which we use to deterministically sample dropout masks from a Bernoulli distribution. These masks are used to zero-out input tensor elements at each network layer. We then reset the RNG using the same initial seed—fixing the drop-out configuration—and continually sample from the DM until we obtain $N$ predictions for that seed. This process is repeated across $M$ different seeds for a total of $M \times N$ predictions. In all experiments, we train and test MC-Dropout with dropout probability $p = 0.3$.

# B  Ablation Studies

## B.1  Sampling Rates

HyperDM provides flexibility at inference time to arbitrarily choose the number of network weights $M$ to sample—analogous to the number of ensemble members in a deep ensemble—and the number of predictions $N$ to generate per sampled weight. In this study, we examine the effect of sampling rates $M, N$ on $\widehat{\text{EU}}$ and $\widehat{\text{AU}}$ on our OOD experiment from Section 5.3.

In our first test, we estimate EU on an OOD measurement for fixed $N = 100$ and variable $M = \{2, 4, 8, 16\}$. Results in Figure 6 indicate that under-sampling weights (i.e., $M \leq 4$) leads to uncertainty maps which underestimate uncertainty around OOD features and overestimate uncertainty around in-distribution features. However, as we continue to sample additional network weights, we observe increased uncertainty in areas around the abnormal feature and suppressed uncertainty around in-distribution features. This result indicates the importance of large ensembles in correctly isolating OOD features from in-distribution features for EU estimation.

In our second test, we repeat the same process but instead fix $M = 10$ and sample $N = \{2, 4, 8, 16\}$ predictions from the DM. Examining the results shown in Figure 7, we observe irregular peaks in the predicted AU at low sampling rates $N \leq 4$. However, as we sample more from the DM and the sample mean converges, $\widehat{\text{AU}}$ becomes more uniformly spread across the entire continental landmass. This result suggests the importance of sampling a large number of predictions for adequately capturing the characteristics of the true likelihood distribution.

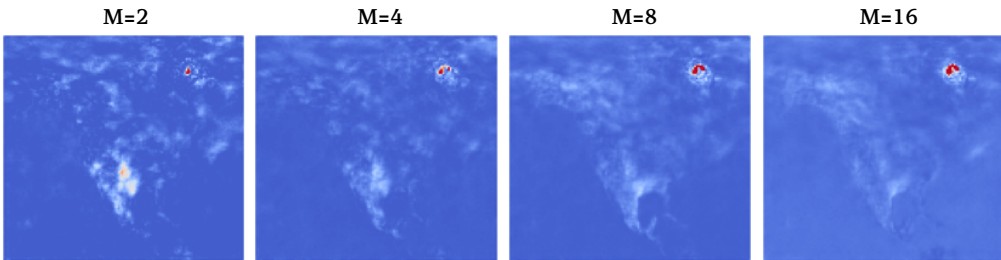

Figure 6: **Effect of sampling more weights on epistemic uncertainty.** As we increase the number $M$ of sampled weights from the hyper-network, uncertainty around out-of-distribution features (i.e., the hot spot in the upper-right) grows and uncertainty around in-distribution features (i.e., everything else in the image) shrinks.

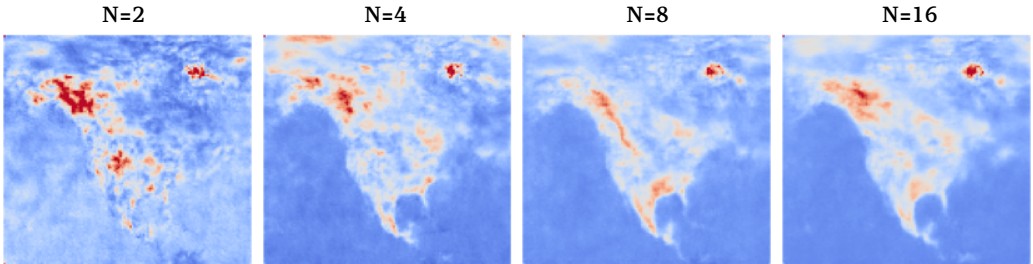

| N=2 | N=4 | N=8 | N=16 |

Figure 7: **Effect of sampling more predictions on aleatoric uncertainty.** As we increase the number $N$ of sampled predictions from the diffusion model, aleatoric uncertainty predictions smooth out more evenly.

Table 4: **Ensemble prediction quality versus BNNs.** When trained on four datasets of various sizes, we observe that HyperDM produces more accurate mean predictions (as indicated by PSNR scores) and higher quality predictive distributions (as indicated by CRPS) than BNNs, except in the extreme low data regime. Highest scores are displayed in **boldface**.

| DATASET SIZE | 100 | 200 | 400 | 800 | 100 | 200 | 400 | 800 |
|---|---|---|---|---|---|---|---|---|
| METHOD | PSNR (DB) ↑ | | | | CRPS ↓ | | | |
| BNN | **10.34** | 11.09 | 13.53 | 13.78 | **0.20** | 0.18 | 0.14 | 0.13 |
| HYPERDM | 8.47 | **18.43** | **20.28** | **20.44** | 0.23 | **0.09** | **0.07** | **0.07** |

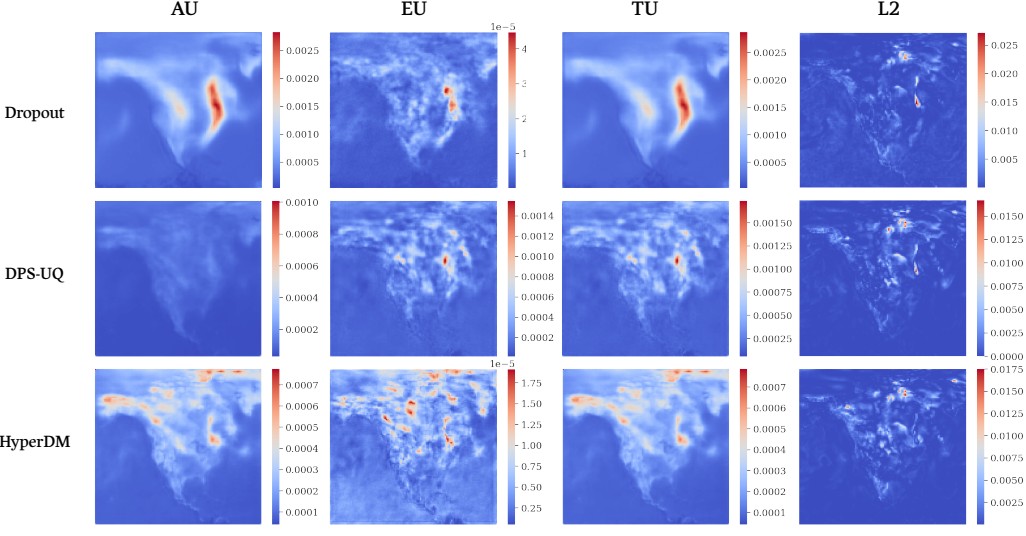

Figure 8: **Uncertainty decomposition on temperature data.** Total uncertainty and its decomposition into epistemic and aleatoric components is shown in the left three columns. The L2 error between the aggregated mean ensemble prediction and the ground truth is shown in the rightmost column. We observe higher total uncertainty around the North American continent, which corresponds with the increased L2 errors around the same areas.

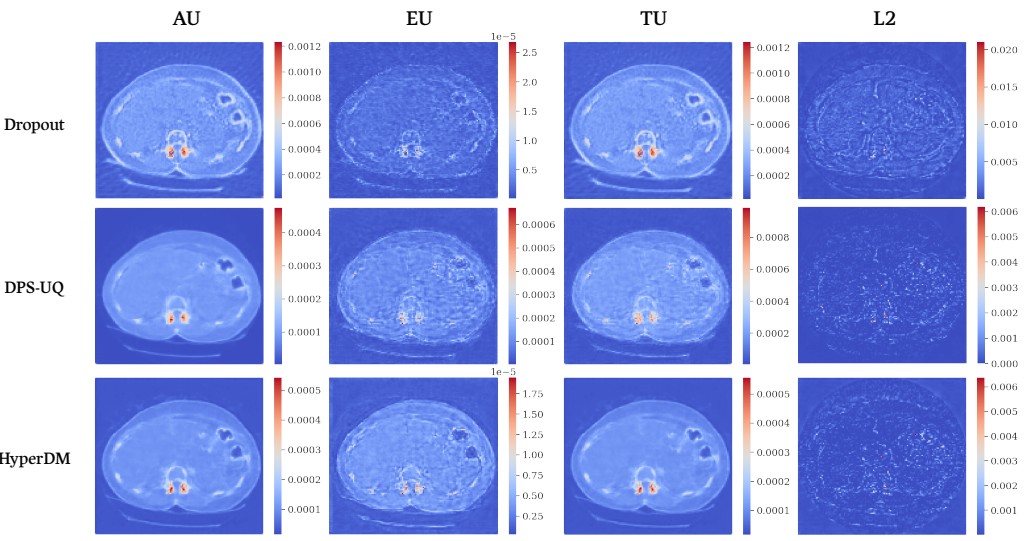

Figure 9: **Uncertainty decomposition on CT data.** Total uncertainty is high near strong features such as the spine and lining of the thoracic cavity, which corresponds to the noisy spots in L2 error map around those areas.

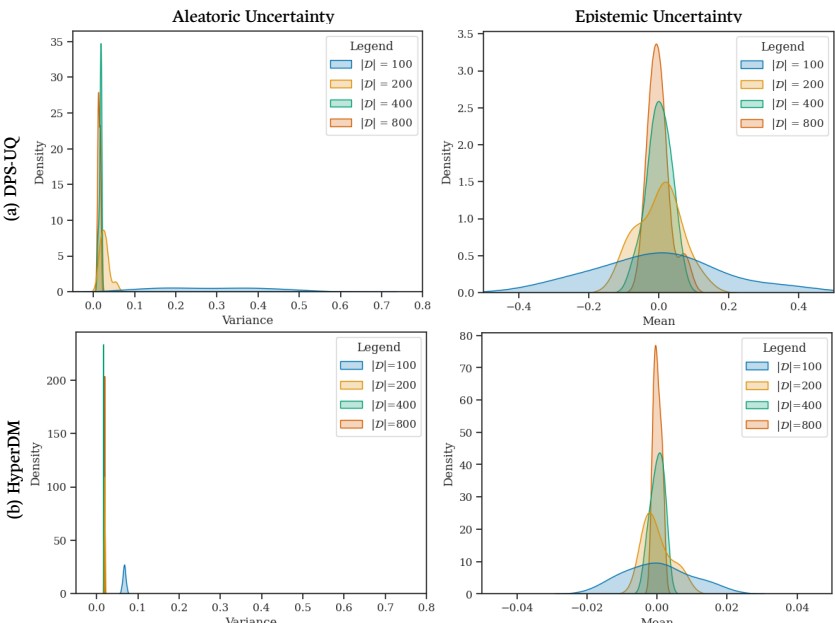

Figure 10: **Varying epistemic uncertainty.** Aleatoric and epistemic uncertainty estimates predicted by (a) DPS-UQ and (b) HyperDM when trained on dataset sizes $|\mathcal{D}| = [100, 200, 400, 800]$.

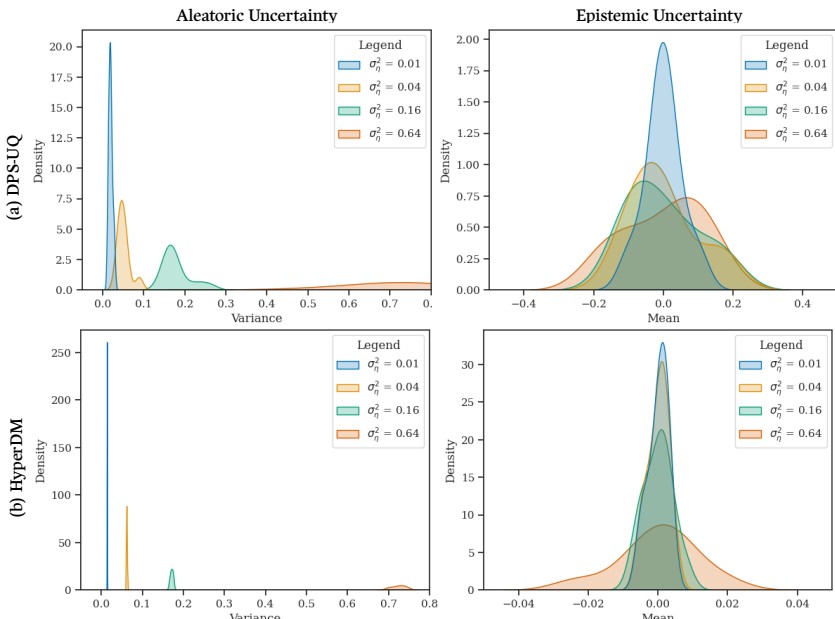

Figure 11: **Varying aleatoric uncertainty.** Aleatoric and epistemic uncertainty estimates predicted by (a) DPS-UQ and (b) HyperDM when trained on noisy datasets $\sigma_\eta^2 = [0.01, 0.04, 0.16, 0.64]$.

