# OpenReview forum: "Estimating Epistemic and Aleatoric Uncertainty with a Single Model"
_NeurIPS.cc/2024/Conference — NeurIPS 2024 poster_

### Official Review · Reviewer_zgME · 2024-07-05

**Soundness:** 3
**Presentation:** 2
**Contribution:** 2
**Rating:** 6
**Confidence:** 4

**Summary:**

The work applies hypernetworks to estimate both aleatoric and epistemic uncertainty in the context of diffusion models. The authors leverage the distribution on weights created by the hypernetwork to estimate uncertainty via the Total Uncertainty (TU) decomposition into Aleatoric Uncertainty (AU) and Epistemic Uncertainty (EU). They validate their method on a toy dataset, lung scan dataset, CT scan dataset and weather predictions.

**Strengths:**

1. The paper is clear and well-written.
2. The authors provide a solution to an important topic, given the prevalence of generative models.
3. The authors apply their method to both a straightforward toy problem and real-life datasets.

**Weaknesses:**

1. The authors seem to be missing a citation where previous researchers have estimated epistemic uncertainty for diffusion models.
    - Berry, Lucas, Axel Brando, and David Meger. "Shedding Light on Large Generative Networks: Estimating Epistemic Uncertainty in Diffusion Models." The 40th Conference on Uncertainty in Artificial Intelligence.
2. There are claims regarding ensembles that appear incorrect. It is not necessary to train $M$ distinct networks; many modern methods work by ensembling certain parts of the network while sharing weights across the rest. Thus, training ensembles is not as expensive as stated.
    - Osband, Ian, et al. "Deep exploration via bootstrapped DQN." Advances in Neural Information Processing Systems 29 (2016).
    - Berry, Lucas, and David Meger. "Normalizing flow ensembles for rich aleatoric and epistemic uncertainty modeling." Proceedings of the AAAI Conference on Artificial Intelligence. Vol. 37. No. 6. 2023.
3. Minor writing comments:
    - The abstract is a bit unusual with two paragraphs and could be shortened.
    - The notation in section 4.2 seems overloaded. Specifically, the subscript on $x$ refers to both the denoising steps and the samples in the training set.

**Questions:**

How does the scalability of hypernetworks compare to that of MC Dropout? In both instances, there are challenges with scaling the processes to networks that require a lot of parameters.

**Limitations:**

The authors address their limitations in the appendix.

---

> ### Author Rebuttal · Authors · 2024-08-07
>
> Thank you for your thoughtful feedback. We have summarized and responded to your questions and concerns below. Please let us know if you have any additional questions or comments and we will do our best to respond promptly.
>
> **1. Missing citation where diffusion models are used to estimate epistemic uncertainty.**
>
> Thank you for bringing this to our attention. We weren’t able to include a citation to DECU [C] in our original submission since it was published after the NeurIPS submission deadline. However, we will certainly update our paper and include a reference and discussion in Section 2.3.
>
> **2. Modern methods ensemble only certain parts of the network, so training ensembles is not necessarily as expensive as stated.**
>
> We agree with the reviewer; this statement should be further clarified. We have attempted to make this point more apparent by adding references to relevant ensembling papers [A, B] in Section 2 and updating our statements in Section 5 to state that our method achieves a reduction in training cost up to $M\times$ that of deep ensembling methods, depending on the number of ensembled parameters.
>
>
> **3. How does scaling of the BHN compare to MC Dropout? Both have issues with scalability.**
>
> In our experiments, we chose to set the BHN just large enough that it produces reasonable weights for the diffusion model. In doing so, we avoid adding a significant increase in the number of trainable parameters. Our BHN incurred only a moderate 10% overhead compared to a baseline diffusion model; from 44.1 min to 48.5 min. In comparison, we observed a slightly smaller training overhead of 6.64% with MC-Dropout;  from 44.1 min to 47.0 min.
>
>
> **4. The subscript on $x$ is overloaded. It refers to both the number of denoising steps and the number of samples in the training set.**
>
> Thank you for pointing this out. We have clarified this notation in the main text by denoting the denoising step $t$ with a superscript and the dataset index $i$ with a subscript. Specifically, a sample $i$ from the dataset at denoising step $t$ is now expressed as $x_i^{(t)}$.
>
> ---
>
> **References**
>
> [A] Osband, Ian, et al. "Deep exploration via bootstrapped DQN." Advances in neural information processing systems 29 (2016).
>
> [B] Berry, Lucas, and David Meger. "Normalizing flow ensembles for rich aleatoric and epistemic uncertainty modeling." Proceedings of the AAAI Conference on Artificial Intelligence. Vol. 37. No. 6. 2023.
>
> [C] Berry, Lucas, Axel Brando, and David Meger. "Shedding Light on Large Generative Networks: Estimating Epistemic Uncertainty in Diffusion Models." The 40th Conference on Uncertainty in Artificial Intelligence.

---

### Official Review · Reviewer_cmte · 2024-07-05

**Soundness:** 2
**Presentation:** 3
**Contribution:** 2
**Rating:** 5
**Confidence:** 4

**Summary:**

The paper proposes a approach, HyperDDPM, to estimate both aleatoric and epistemic uncertainty with an ability to disentangle them. This result is achieved by combining diffusion networks (DDPM) to sample predictions from single set of model weights, and Bayesian Hyper Network to generate sets of weights for main DDPM model. Using synthetic data the authors show, that proposed approach can estimate aleatoric and epistemic components of predictive uncertainty, and provide evidence for real-world applicability on medical imaging and weather forecasting tasks.

**Strengths:**

1. The paper presents a new and potentially useful way of producing ensembled predictions.
2. On synthetic data it shows that this approach is sensitive to both epistemic and aleatoric uncertainty components when ground truth influence of such components is isolated.
3. It shows on real world tasks, like medical image reconstruction and meteorological forecasting, that proposed architecture can outperform some, and perform on par with the other baseline methods.
4. The authors provide codebase used to train and evaluate models, which allows for better transparency and reproducibility.

[After rebuttal comment] I increased my score from 3 to 5. I am still not convinced that uncertainty is treated in coherent way as the paper jumps from the Bayesian predictive distribution to the law of total variance without any connection betweeen them. Additionally, real world experiments do not evaluate uncertainty numerically.

**Weaknesses:**

Here I summarize weaknesses of the paper, some details on the weaknesses are clarified by the Questions section of the review.
0. The paper lacks a Contributions section, clearly and concisely stating which claims and results within the paper are new.
1. The experimental section does not provide sufficient evidence of the ability of the proposed approach to disentangle epistemic and aleatoric components of uncertainty.
2. Theoretic justifications do not fully align nicely with the engineering approach taken for experiments, and contain questionable claims.
3. Some details on training procedure are unclear.
4. The paper lacks a straightforward comparison of computational overhead of baselines and proposed method, during training and evaluation. Some of these are present scattered within paper text, but the paper would benefit from having them collected in one place, given that both baselines and proposed method incur significant overhead.

**Questions:**

1. In explanation of eq. 1 the authors claim that likelihood under the integration contains information about aleatoric uncertainty, which can be argued as true, but it seems that marginal distribution on the left side of the equation should provide a better estimate of aleatoric component, given that it's marginalized over model weights distribution, and thus epistemic component is eliminated. This can be further confirmed by the way the authors approach estimating aleatoric uncertainty in practice, described in eq. 10 - here the authors take expectation over the model weights in the outer operator.

2. In Appendix A authors describe training of BHN as:
	- compute an L2 loss, and back-propagate the gradients to update the hyper-network weights.
   But in the next paragraph the following is said:
    - The BHN is trained with the same L2 loss and an additional Kullback-Leibler (KL) divergence loss

   How exactly was BHN training performed? It is also quite interesting why BHN training did not collapse to an optimal mode with very low variance over produced model weights. How often was input noise vector for BHN changed during training? On per-batch level, per-input, or using some other strategy?

3. Toy experiment on synthetic data is only performed to estimate AU and EU when corresponding uncertainty component is changing. It seems to not fully support claim that proposed approach can disentangle AU from EU. It would be much more demonstrative of claim, if the experiment was done by independently varying noise level and train dataset size, and plotting results on AU and EU axes.

4. Real world experiments only show base performance of proposed architecture, but do not provide reasonably in-depth analysis of UE capabilities on such task, apart from figures with uncertainty maps over generated images. The paper would benefit greatly from some form of analysis of how uncertainty estimates improve prediction/reconstruction quality when filtering out low-confidence inputs (i.e. rejection verification).

5. Was training of base model of MC-Dropout baseline performed using active dropout masks in the same way as during inference? Some degree of performance hit that MC-Dropout model suffered could be explained by difference between training and evaluation dropout regimes.

**Limitations:**

The authors have adequately addressed limitations of their work.

---

> ### Author Rebuttal · Authors · 2024-08-07
>
> Thank you for your thoughtful feedback. We have summarized and responded to your questions and concerns below. Please let us know if you have any additional questions or comments and we will do our best to respond promptly.
>
> **1. The paper doesn’t have a dedicated contributions section.**
>
> Thank you for the suggestion. We have described the technical novelty and specific contributions of our paper in bullet #2 of our global response.
>
> **2. The paper lacks a straightforward comparison of computational overhead of baselines and the proposed method during training and evaluation.**
>
> We have added a table in our global response that shows the computational runtime required by all methods for training and evaluation. This result will be added to the main paper.
>
> **3. In Equation 1, the marginal distribution on the left side of the equation should provide a better estimate of the aleatoric component, given that it's marginalized over model weights distribution. This can be further confirmed by the way the authors approach estimating aleatoric uncertainty in practice, described in Equation 10.**
>
> We acknowledge that future work should explore a better estimate of aleatoric uncertainty in Equation 1 and will mention this in the conclusion. However, in the main paper, we derive Equation 1 from Ekmekci et al. [9], which is supported by numerous works [24, A, B], in which the predictive distribution contains both an aleatoric and epistemic component. Likewise, our derivation for Equation 10 stems from prior works [49, C, D] which apply the law of total variance to disentangle epistemic and aleatoric uncertainties.
>
>
> **4. In Appendix A, authors describe training of BHN as: compute an L2 loss, and back-propagate the gradients to update the hyper-network weights. But in the next paragraph the following is said: "The BHN is trained with the same L2 loss and an additional Kullback-Leibler (KL) divergence loss."**
>
> We apologize for the typo in Appendix A, which has been corrected in the text: The Bayesian neural network (BNN), not Bayesian hyper-network (BHN), is trained with an additional KL divergence loss.
>
> **5. How is BHN training performed? How often is the noise vector changed during training (e.g., per-input, per-batch, etc.)? Why did the BHN not collapse to a small optimal mode?**
>
> The BHN is trained only with an L2 loss. During training, the noise vector is changed per-batch, and for the majority of our experiments we set the batch size to 32. The BHN doesn’t collapse to a mode because there are a variety of network weights that yield plausible predictions with respect to L2 distance.
>
> **6. The toy experiment should be done by independently varying noise level and training dataset size and plotting results on AU and EU axes.**
>
> This is a great suggestion. We  have provided these figures in the general response PDF, which show that our method performs similarly to deep ensembles. We note, however, both methods struggle to fully disentangle EU and AU when one type of uncertainty is very high. In Figure R1, both DPS-UQ and HyperDDPM estimate a similar AU across different dataset sizes, indicating that the AU estimate is relatively unaffected by the increasing EU of the problem. Similarly, in Figure R2, both methods estimate a similar EU across different noise levels, except in the extreme noise scenario. Prior work [E] suggests that one should disregard the AU estimate when EU is high (i.e., the measurement is out-of-distribution). We will incorporate these figures and corresponding descriptions into the main text.
>
> **7. The paper would benefit from a more in-depth analysis on real world problems. For example, how can EU help improve reconstruction quality (e.g., by filtering out low-confidence outputs / rejection verification)?**
>
> We will update the main text to provide more concrete examples of use cases for EU and AU estimation (e.g., active learning for weather station placement, predicting long-tailed events with high EU, and cost mitigation strategies). We note that our current experimental results demonstrate that our approach can be used for rejection verification. For instance, in both real-world experiments, we provide in-distribution and out-of-distribution measurements to the model, and we are able to identify which parts of the prediction are unreliable and should be investigated more thoroughly by a trained expert.
>
>
> **8. Were the drop-out masks kept consistent during training and evaluation of the MC Dropout baseline? Any differences could explain the performance drop.**
>
> We generate random activation masks with the same drop-out probability $p=0.3$ for both training and evaluation. During training, the activation masks are randomly generated per-batch. At evaluation time, we use the same procedure to randomly generate $M=10$ masks and compute AU and EU according to Equations 10 and 11. We will clarify this procedure in the revised paper.
>
> ---
> **References**
>
> [A] Fellaji, Mohammed, and Frédéric Pennerath. "The Epistemic Uncertainty Hole: an issue of Bayesian Neural Networks." arXiv preprint arXiv:2407.01985 (2024).
>
> [B] Kwon, Yongchan, et al. "Uncertainty quantification using Bayesian neural networks in classification: Application to biomedical image segmentation." Computational Statistics & Data Analysis 142 (2020): 106816.
>
> [C] Schreck, John S., et al. "Evidential deep learning: Enhancing predictive uncertainty estimation for earth system science applications." arXiv preprint arXiv:2309.13207 (2023).
>
> [D] Joshi, Shalmali, Sonali Parbhoo, and Finale Doshi-Velez. "Pre-emptive learning-to-defer for sequential medical decision-making under uncertainty." arXiv preprint arXiv:2109.06312 (2021).
>
> [E] Mukhoti, Jishnu, et al. "Deep deterministic uncertainty: A new simple baseline." Proceedings of the IEEE/CVF Conference on Computer Vision and Pattern Recognition. 2023.

---

> > ### Comment · Reviewer_cmte · 2024-08-14
> > **Rebuttal well received**
> >
> > Dear authors,
> >
> > your rebuttal was well received and you partially addressed my concerns. I will decide on the score changes after the discussion with other reviewers.

---

### Official Review · Reviewer_YZoM · 2024-07-07

**Soundness:** 3
**Presentation:** 4
**Contribution:** 3
**Rating:** 6
**Confidence:** 4

**Summary:**

This paper proposes HyperDDPM, a novel uncertainty quantification method for generative tasks that uses a hypernetwork and a denoising diffusion probabilistic model (DDPM) and outputs both aleatoric and epistemic uncertainty estimates. HyperDDPM is evaluated on a toy task with available ground-truth uncertainties as well as two real-life tasks, CT reconstruction and weather forecasting. The authors find that HyperDDPM gives accurate uncertainty estimates that are on par with deep ensembles.

**Strengths:**

- Uncertainty quantification in generative models is a high-impact research area.
- The paper is easy to read.
- HyperDDPM is a single-model uncertainty estimator that performs on par with ensembles that are SotA in many uncertainty application domains.
- The use of hypernetworks for uncertainty quantification is an interesting avenue.
- The experiments are extensive and clearly explained. The toy experiment allows for fine-grained control and comparison with ground-truth quantities, whereas the real-world tasks of CT reconstruction and weather forecasting are important application areas. The authors provide both quantitative and qualitative evaluation.

**Weaknesses:**

- The paragraph starting at L29 is (i) misleading and (ii) vague. (i) A useful uncertainty estimate does not necessarily have to differentiate between aleatoric and epistemic uncertainty. A total/predictive uncertainty estimate subsumes both of these sources of uncertainty and can be excellent at predicting the underlying model's correctness on an input $x$ (e.g., see Fig. 6 in (Mukhoti et al., 2023; CVPR)). (ii) The "valuable insights into the strengths and weaknesses of a predictive model" are not elaborated, neither are the pathways these insights offer "towards improving [the model's] performance". I _do_ agree that uncertainty disentanglement is an important and interesting research direction but this paragraph doesn't quite answer _why_. Consider providing concrete use cases for the separate aleatoric and epistemic estimates and how they can improve the model's performance.
- A more complex model class generally does not lead to a decrease of epistemic uncertainty (Section 3.2). Larger models introduce more degrees of freedom into the learning problem, leading to a larger set of models consistent with the training data (as captured by the Bayesian posterior $p(\phi \mid \mathcal{D})$).
- The claim that "This estimate converges to the true AU of the inverse problem as $N \to \infty$" is imprecise and unproven. The statement assumes that (i) the score estimator $s_\phi$ is expressive enough to model the true score of the generative process perfectly, (ii) $s_\phi$ is trained on the entire data distribution (i.e., the expectation in Eq. (5) is optimized), and (iii) the global minimum is reached during optimization. Only then can one argue that the true AU of the inverse problem is reached at an _in-distribution_ input $x$ when one samples infinitely many samples from the implicit distribution.

**Minor Issues:**
- Consider making the figures vector-graphical. The current PNGs are quite pixelated.
- The precise formulation of Eq. (7) would be $\mathbb{E}_{(x, y) \sim \mathcal{D}}\left[\mathbb{E}_{z \sim \mathcal{N}(0, \sigma^2)}\left[\mathcal{L}(f(y \mid \phi(z)), x)\right]\right]$. The sampling is not captured by the original formula.
- Consider moving Appendix C into the main paper in the final version.

**Questions:**

- Why should one trust their aleatoric uncertainty estimates on OOD data (Fig. 2)? Usually, a two-step procedure is carried out (see Fig. 3 in Mukhoti et al., 2023; CVPR) where the AU estimates are only considered if the EU estimates are below a certain threshold. It also seems that at OOD regions, the AU and EU estimates are highly correlated.

**Limitations:**

The authors address the limitations of their work adequately.

---

> ### Author Rebuttal · Authors · 2024-08-07
>
> Thank you for your thoughtful feedback. We have summarized and responded to your questions and concerns below. Please let us know if you have any additional questions or comments and we will do our best to respond promptly.
>
> **1. The paragraph starting at L29 is misleading and vague. A useful uncertainty estimate does not necessarily have to differentiate between aleatoric and epistemic uncertainty. Also, consider providing concrete use cases for the separate aleatoric and epistemic estimates and how they can improve the model's performance.**
>
> This statement was indeed too strong. We have changed “To be useful…” to “To be most useful…” and added the following motivating text to the paper: “In applications such as weather forecasting, epistemic uncertainty can be used to inform the optimal placement of new weather stations. Additionally, in medical imaging, decomposition of uncertainty into its aleatoric and epistemic components is important for identifying out-of-distribution measurements where model predictions should be verified by trained experts.”
>
> **2. More complex models introduce more degrees of freedom into the learning problem, leading to a larger set of models consistent with the training data and an increase, not decrease, of epistemic uncertainty.**
>
> We agree with the reviewer’s statement and will remove the referenced statement from the main text.
>
>
> **3. The claim that the “estimate converges to the true aleatoric AU of the inverse problem” is imprecise. This is only true under the assumption that (i) the score estimator is expressive enough to model the true score of the generative process perfectly, (ii) is trained on the entire data distribution, and (iii) the global minimum is reached during optimization.**
>
> Thank you for pointing this out. We will specify the exact assumptions under which our statement is true, as listed by the reviewer.
>
> **4. Why should we trust AU estimates when EU is high? Prior work suggests that we should only consider AU when EU is below a threshold.**
>
> Our results illustrate that our single-model ensembling approach produces similar estimates of AU and EU as multi-model deep ensembling approaches. It's true$\textemdash$and important to acknowledge$\textemdash$that AU estimates are only meaningful when there's sufficient data that EU estimates are low. We will highlight this point in the revision.
>
> **5. The precise formulation of Eq. (7) would be $\mathbb{E}\_{(x, y) \sim \mathcal{D}}\left[\mathbb{E}\_{z \sim \mathcal{N}(0, \sigma^2)}\left[\mathcal{L}(f(y \mid \phi(z)), x)\right]\right]$. The sampling is not captured by the original formula.**
>
> Thank you for the correction. As suggested, we have updated the formula in Equation 7 to more accurately denote the sampling of noise vectors $z\sim\mathcal{N}(0,\sigma^2)$.
>
> **6. Move Appendix C into the main paper for the final version.**
>
> Thanks for the suggestion. We have integrated Appendix C into the revised paper.
>
>
> **7. Make figures vector-graphical.**
>
> Thank you for the suggestion. We have updated all figures in the revision to use vector graphics.

---

### Official Review · Reviewer_pwCt · 2024-07-14

**Soundness:** 3
**Presentation:** 3
**Contribution:** 2
**Rating:** 6
**Confidence:** 3

**Summary:**

Diffusion has been applied in many domains beyond image generation, e.g. weather forecasting and CT reconstruction. Many of these applications are safety-critical as such the model should be capable of expressing uncertainty. As such the submission proposes a method based on the idea of hypernet for uncertainty quantification: The method would train a network that maps from noise to the a set of weights of the U-net of the diffusion, then generated weight is then treated a sample from the "posterior" and later used to estimate the epistemic uncertainty. In order to estimate aleatoric uncertainty, the method uses MC estimation to estimate var(y | x, phi) for a fixed model weight and then average the variance across all weights generated by the hypernet. The paper then evaluates the method on weather forecasting problem and CT image reconstruction problem: In both setting, the method demonstrates a nice decomposition of aleatoric and epistemic uncertainty.

Overall I find the application interesting and paper is nicely written and presentation is clear. However I find the technical contribution a little limited although the empirical results seem to be promising.

**Strengths:**

- The problem studied, uncertainty quantification for diffusion, is important.

- The method proposed is technical sound and it works well in empirical evaluation.

- The experiment and hyper-parameter setting are clearly presented in the appendix.

**Weaknesses:**

- Error bar / standard deviation is not provided.

- It would be nice if a summarization of runtime comparison can be provided, i.e. total training time, total inference time.

- The technical novelty is not very outstanding (though the application seems to be novel), see the bullet point below for more detail.

- The contribution of the paper is not described in a very precise way: From my perspective, the paper applies existing methods, i.e. Bayesian hyper-network, in a new setting: **diffusion models**. However this is not highlighted in the abstract. The abstract gives readers the impression that the paper is proposing uncertainty quantification methods for generic problems.

- (Minor) Lack of baseline methods: Some strong BNN baselines such cyclical SGLD or linearized Laplace, SWAG are not included. MC-dropout used in the paper is a pretty out-of-date technique for uncertainty quantification problem in my opinion.

- (Minor) In Fig. 1, hyper-network is written as hypernetwork in the figure body (a), it would be better if the term can be spelled in a consistent way.

**Questions:**

- What's the training overhead for the hyper-network?

- How should one set the complexity of the hyper network? If the network is too large / small, would the uncertainty vanish?

- How does the number of samples, i.e. M and N, affect the performance?

- What does BNN mean in the main text? Does it refer to MC-dropout or DPS-UQ or both?

**Limitations:**

- To estimate epistemic uncertainty, although the method only needs a single model (in contrast to BNNs that requires multiple copies/samples of a model), the method still needs **multiple forward propagation** to estimate the uncertainty, so the part of the computational cost at test time still exists

---

> ### Author Rebuttal · Authors · 2024-08-07
>
> Thank you for your thoughtful feedback. We have summarized and responded to your questions and concerns below. Please let us know if you have any additional questions or comments and we will do our best to respond promptly.
>
> **1. Provide error bars / standard deviations.**
>
> We have trained five sets of ensembles and computed the variance across the EU and AU estimates. The uncertainty estimates are quite consistent, but due to space constraints we are not able to fit these figures in our response. We will include them in the revised text. We have also provided RMSE errors and variances (i.e., total uncertainty) for our experimental results in Figures 8 and 9 of the supplement.
>
> **2. Provide a summary figure comparing the computational overhead of baselines and the proposed method during training and evaluation.**
>
> We have added a figure in our global response that shows the computational runtime required by all methods for training and evaluation.
>
> **3. The technical novelty of the paper is not described precisely. The abstract should highlight the application of existing methods in a new setting.**
>
> Thank you for the suggestion. We have described the technical novelty and specific contributions of our paper in Question 2 of the global response.
>
>
> **4. What's the training overhead for the hyper-network? How should one set the complexity of the hyper network? If the network is too large / small, would the uncertainty vanish?**
>
> We chose to set the hyper-network just large enough that it produces reasonable weights for the diffusion model. In doing so, we avoid adding a significant increase in the number of trainable parameters. Specifically, this amounted to around a 10.05% increase in training time (i.e., from 44.1 min to 48.5 min) for our weather forecasting and CT reconstruction experiments. If the hyper-network architecture is too small, we found that it outputs poor quality weights for the diffusion model. This text will be incorporated into the revised paper.
>
> **5. How does the number of samples, i.e. M and N, affect performance?**
>
> In general, we found that under-sampling M (i.e., the number of samples from the implicit weight distribution) leads to uncertainty maps which underestimate uncertainty around out-of-distribution features and overestimate uncertainty around in-distribution features. As we continually sample network weights, we observe increased uncertainty in areas around the abnormal feature and suppressed uncertainty around in-distribution features. Similarly, under-sampling N (i.e., the number of samples from the likelihood) leads to irregular peaks in the predicted aleatoric uncertainty maps. As we sample more from the diffusion model and the sample mean converges, the aleatoric uncertainty becomes more uniform. Please see our ablation study in Section B.1 of the supplement for additional figures and details.
>
> **6. What does BNN mean in the main text? Does it refer to MC-dropout or DPS-UQ or both?**
>
> BNN stands for Bayesian neural network. This acronym is specified in Section 2.1, and the network model for the BNN used in our experiment is described in Section A of the supplement. We have attempted to clarify this by placing a reference in the toy experiment which points to the definition of BNNs in Section 2.1.
>
> **7. Provide stronger BNN baselines (e.g., cyclical SGLD, linearized Laplace, SWAG).**
>
> In our non-toy experiments, we did not include a BNN baseline because we wanted to focus on evaluating our technique against methods which are able to predict both AU and EU for a more direct comparison. We will include SQR [B] and Kendall et al. [A] as additional baselines in our benchmark.
>
>
> **8. In Figure 1, hyper-network is written as “hypernetwork” in the figure body. The term should be spelled in a consistent way.**
>
> Thank you for bringing this to our attention. We have corrected the text in Figure 1 to more consistently reflect the spelling of the term “hyper-network” throughout the paper.
>
> ---
>
> **References**
>
> [A] Kendall, Alex, and Yarin Gal. "What uncertainties do we need in bayesian deep learning for computer vision?." Advances in neural information processing systems 30 (2017).
>
> [B] Tagasovska, Natasa, and David Lopez-Paz. "Single-model uncertainties for deep learning." Advances in neural information processing systems 32 (2019).

---

> > ### Comment · Reviewer_pwCt · 2024-08-13
> >
> > I would like to thank the author for the response, I kept my score unchanged.
> >
> > Overall, I did not find the method itself very technically novel and did not find the baseline method (MC Dropout) considered very satisfying (note that many modern networks even don't have dropout!).
> >
> > However, the application itself is novel, and does not seem to have been studied often in existing literatures, so I still think this work can be valuable to the community, as long as the authors state clearly the contribution.

---

### Author Rebuttal · Authors · 2024-08-06

We thank the reviewers for their thoughtful and constructive feedback. We are glad they found our proposed method novel (YZoM, cmte), performant against the current state-of-the-art (pwCt, YZoM, cmte), easy-to-understand (pwCt, YZoM, zgME), and well-supported by experiments (pwCt, YZoM). We have summarized and responded to common questions and concerns below.

**1. (pwCt, cmte) Provide a summary figure comparing the training and inference times of the proposed method and baselines.**

The attached PDF provides a summary figure which compares the training and inference times required by our method, MC dropout, and DPS-UQ. We will add this table to the revised text.

**2. (pwCt, cmte) Specify contributions explicitly / provide a contributions section.**

The technical novelty of our paper is application of Bayesian hyper-networks in a novel setting (i.e., diffusion models) to estimate both epistemic and aleatoric uncertainties from a single model. The abstract has been revised as follows: “estimate epistemic and aleatoric uncertainty with a single model” → “estimate epistemic and aleatoric uncertainty with a single diffusion model.” Furthermore, our contributions can be summarized as follows (this text will be added as a contribution section to the manuscript):
- We apply Bayesian hyper-networks in a novel setting, diffusion models, to estimate both epistemic and aleatoric uncertainties from a single model.
- We conduct a toy experiment with ground truth uncertainties and show that the proposed method accurately predicts both sources of uncertainty.
- We apply the proposed method on two mission-critical real-world tasks, CT reconstruction and weather forecasting, and demonstrate that our method achieves a significantly lower training overhead and better reconstruction quality compared to existing methods.
- We conduct ablation studies investigating the effects of ensemble size and the number of ensemble predictions on uncertainty quality, which show (i) that larger ensembles improve out-of-distribution detection and (ii) that additional predictions smooth out irregularities in the aleatoric uncertainty estimates.

---

### Decision · Program_Chairs · 2024-09-25

**Decision:**

Accept (poster)

**Comment:**

This paper proposes a Hyper-network approach that applied to diffusion models. By leveraging the sampling ability of diffusion model, the proposed approach claims to separate the epistemic and aleatoric uncertainty. Although the technical contribution is limited, it shows some preliminary results. The paper will be stronger if more evidence is shown.